# mTOR Inhibitors in Advanced Biliary Tract Cancers

**DOI:** 10.3390/ijms20030500

**Published:** 2019-01-24

**Authors:** Chao-En Wu, Ming-Huang Chen, Chun-Nan Yeh

**Affiliations:** 1Division of Hematology-Oncology, Department of Internal Medicine, Chang Gung Memorial Hospital, Linkou branch, Chang Gung University, Taoyuan 333, Taiwan; jiaoen@gmail.com; 2School of Medicine, National Yang-Ming University, Taipei 112, Taiwan; mhchen9@gmail.com; 3Department of Oncology, Taipei Veterans General Hospital, Taipei 112, Taiwan; 4Department of General Surgery and Liver Research Center, Chang Gung Memorial Hospital, Linkou branch, Chang Gung University, Taoyuan 333, Taiwan

**Keywords:** mTOR, advanced biliary tract cancers

## Abstract

Patients with advanced biliary tract cancers (BTCs), including cholangiocarcinoma (CCA), have poor prognosis so novel treatment is warranted for advanced BTC. In current review, we discuss the limitations of current treatment in BTC, the importance of mTOR signalling in BTC, and the possible role of mTOR inhibitors as a future treatment in BTC. Chemotherapy with gemcitabine-based chemotherapy is still the standard of care and no targeted therapy has been established in advanced BTC. PI3K/AKT/mTOR signaling pathway linking to several other pathways and networks regulates cancer proliferation and progression. Emerging evidences reveal mTOR activation is associated with tumorigenesis and drug-resistance in BTC. Rapalogs, such as sirolimus and everolimus, partially inhibit mTOR complex 1 (mTORC1) and exhibit anti-cancer activity in vitro and in vivo in BTC. Rapalogs in clinical trials demonstrate some activity in patients with advanced BTC. New-generation mTOR inhibitors against ATP-binding pocket inhibit both TORC1 and TORC2 and demonstrate more potent anti-tumor effects in vitro and in vivo, however, prospective clinical trials are warranted to prove its efficacy in patients with advanced BTC.

## 1. Introduction of Bile Duct Cancers

Bile duct cancers (BTCs) including intrahepatic/extrahepatic cholangiocarcinoma (CCA), gallbladder cancer (GBC), and Ampullar Vater cancer, are the malignant neoplasms arising from epithelial cells of bile ducts [1]. CCA was considered as primary liver cancer and, currently, the term CCA has been used for bile duct cancers arising from intrahepatic and extrahepatic bile system, excluding the malignancies of gallbladder and Ampulla of Vater.

The estimated annual cases of primary liver cancers including intrahepatic CAA is 42,220 in the United States [2] and around 15% of them are intrahepatic CCA according to Surveillance, Epidemiology, and End Results (SEER) program [3,4]. Estimated 12,190 cases of gallbladder and other biliary cancers are diagnosed annually in the United States [2]. Although they are uncommon and relatively rare, the patients with BTC have a poor prognosis because most of them are locally advanced at presentation and high recurrence rate for the early stage after curative surgery [5,6]. The efficacy of systemic treatment is limited, therefore, novel agents are warranted for the patients with biliary tract cancers.

## 2. Current Evidences of Systemic Treatment for Advanced Bile Duct Cancers

### 2.1. In the Era of Chemotherapy

Systemic chemotherapy is the standard treatment in biliary duct cancers based on a randomized study which showed fluorouracil (FU)-based systemic chemotherapy provided longer overall survival (6 versus 2.5 months) than best supportive care alone in 90 eligible patients with pancreatic (*n* = 53) or biliary cancer (*n* = 37) [7]. Therefore, chemotherapy with FU-based regimens proved the efficacy of chemotherapy and became the standard of care for patients with advanced BTC in 1996. A later study in patients with advanced pancreatic cancer showed gemcitabine-treated patients experienced better clinical benefit response compositing of measurements of pain, Karnofsky performance status, and body weight (23.8% vs. 4.8%, *p* = 0.0022) and longer overall survival (OS, 5.65 and 4.41 months, *p* = 0.0025) than 5-FU-treated patients [8], gemcitabine was also wildly used in patients with advanced BTC. Subsequently, chemotherapy with FU, and gemcitabine, with or without platinum has been studied, but the optimal chemotherapy regimen has been debated for more than a decade.

In 2007, pooled phase II studies by Eckel et al. showed superior response rates (RRs) and tumor control rates (TCRs) of gemcitabine- or platinum-based regimens and highest RRs and TCRs was found in the gemcitabine/platinum combination subgroup so this study concluded that gemcitabine/platinum combination represented the provisional standard for chemotherapy [9] even lack of direct comparison of gemcitabine and 5-FU in these patients. In 2010, ABC-02 trial, the first randomized phase III study in advanced BTC, reported that gemcitabine plus cisplatin has better TCRs (81.4% versus 71.8%, *p* = 0.049), median progression-free survival (PFS, 8.0 months versus 5.0 months, *p* < 0.001) and median OS (11.7 months versus 8.1 months, *p* < 0.001) than gemcitabine alone [10] so the combination of gemcitabine and cisplatin has been considered the standard of care as the first-line treatment in patients with advanced BTC and widely used in clinical practice [11]. This regimen has not been compared head to head with other gemcitabine-based combinations except gemcitabine plus TS-1 which demonstrated non-inferiority in the Japanese phase III FUGA-BT study [12]. This study enrolled a total of 354 patients with chemotherapy-naïve advanced BTC and a preliminary report presented at the 2018 American Society of Clinical Oncology (ASCO) Gastrointestinal Cancers Symposium showed the combination of gemcitabine/TS-1 was non-inferior in terms of median OS (15.1 versus 13.4 months), median PFS (6.8 versus 5.8 months), and objective RRs (30% versus 32%) so that this combination can be considered as another standard treatment in patients with advanced BTC.

### 2.2. Development of Targeted Therapy in Advanced BTC

Few prospective trials have been undertaken of first-line chemotherapy and targeted therapy in advanced BTC. Molecularly targeted agents targeting vascular endothelial growth factor (VEGF) or epidermal growth factor receptor (EGFR) were investigated in advanced BTC. Although the addition of bevacizumab [13] or cetuximab [14] to chemotherapy showed promising clinical results in phase II trials, randomized study [15,16] failed to demonstrate additional activity of cetuximab when it combined with chemotherapy. In a study of pooled trials published during January 2000 to January 2014, the authors concluded that triplet combinations of gemcitabine/FU/platinum and gemcitabine-based chemotherapy plus targeted therapy (predominantly targeting EGFR) are most effective concerning TCRs and survivals [17]. However, gemcitabine-based chemotherapy is still the standard of care in advanced BTC and the use of additional targeted therapy is questionable.

### 2.3. Immune Checkpoints Inhibitors

The immune checkpoints inhibitors against cytotoxic T-lymphocyte-associated protein 4 (CTLA-4), programmed cee death protein-1(PD-1), or programmed death-ligand 1 (PD-Ll) have been developed to show efficacy in a variety of cancers. Nakamura et al. found that the poorest prognosis for BTC patients was in those with significant enrichment of hypermutated tumors and elevated expression of immune checkpoint molecules such as CTLA-4 and IDO but which are associated with favourable clinical response to anti-PD-L1 treatment [18]. In this study, 45.2% of patients showed an increase in the expression of immune checkpoint molecules. In Keynote-026 (NCT02054806), a phase 1b trial to evaluate the safety and efficacy of pembrolizumab in advanced pre-treated BTC patients, Bang et al. [19] reported interim results that 8 out of 23 PD-L1-positive patients (35%) had PD and SD and some of them had disease control lasting for 40+ weeks. A number of immunotherapy studies are currently recruiting and ongoing [20].

In addition, based on data from the patients with microsatellite instability-high (MSI-H) or deficient mismatch repair (dMMR) cancers enrolled across uncontrolled, multi-cohort, multi-center, single-arm clinical trials, in May 2017, the US FDA approved pembrolizumab for treatment of a variety of advanced MSI-H or dMMR solid tumors (including BTC) [21] so the patients with advanced BTC harboring MSI-H or dMMR are candidates for immune checkpoint inhibitors.

## 3. Molecular Alterations in Cholangiocarcinoma

A variety of molecular alterations involving both oncogenes (e.g., *RAS* [22,23,24], *BRAF* [25], *ERBB2/HER2*, *EGFR* [26], and *PIK3CA* (phosphoinositide 3-kinase, catalytic, α-polypeptide) [27]) and tumor suppressor genes (e.g., *p53* [23], *SMAD4* [28], and *CDKN2A* [29]) have been described in invasive BTC [30]. Most of the genetic alterations involve phosphoinositide 3-kinase (PI3K)/AKT/mammalian target of rapamycin (mTOR) through MAPKinase activation or p53 suppression resulting in activation of mTOR. p53 negatively regulates the PI3K/AKT/mTOR pathway via its upregulation of phosphatase and tensin homolog (PTEN), TSC2, AMP-activated protein kinase (AMPK), and other proteins [31].

In addition, gene expression profiling of BTC compared with normal biliary epithelium has identified upregulated *ribosomal protein S6 kinase*, *70kD* (*RPS6K* encoding p70-S6K) and *eukaryotic translation initiation factor 4E* (*EIF4E*), which are two important downstream mediators of AKT/mTOR signaling pathway, as well as the potential drug target insulin-like growth factor 1 receptor (IGF1-R) [32]. The collective evidences of genetic studies in BTC detailed above suggest mTOR plays a central and critical role in invasive BTC, therefore, targeting mTOR pathway by mTOR inhibitors could be envisioned as a novel treatment in advanced BTC (Figure 1).

## 4. mTOR Pathway in Cancers

### 4.1. mTOR, Its Complexes and Downstream Regulations in Cancers

The serine/threonine kinase mTOR, a member in a family of protein kinase called *PI3K*-related kinases, integrates intracellular and extracellular signal transduction leading to regulation of in a variety of cellular functions such as cell cycle progression, cell metabolism, cell proliferation, survival [33,34,35,36]. The mTOR pathway is dysregulated in various cancers including cholangiocarcinoma [37,38], making mTOR an important target for the development of new anticancer drugs [39,40].

The mTOR exists in two structurally and functionally distinct complexes, mTOR complex 1 (mTORC1) and mTOR complex 2 (mTORC2), which regulated by and regulate distinct signaling pathways resulting from different complex co-factors. Both complexes contain mTOR and a protein, called mLST8, that associates with its kinase domain. It is considered that the functional differences between mTORC1 and mTORC2 result from the other core components such as Raptor in mTORC1 and a complex of Rictor and mSIN1 in mTORC2.

mTORC1 is the downstream of the two proto-oncogenes kinase pathways, PI3K/AKT as well as RAS/RAF/MEK/ERK, through inhibition of TSC2 and PRAS40, both are negative regulators of mTORC1 [41,42,43,44,45,46]. mTORC1 is the upstream of two distinct pathways which control translation of specific subsets of mRNA. One involves p70-S6K, and another pathway is related with eukaryotic initiation factor 4E binding protein-1 (4E-BP1) [47]. The PI3K/AKT/mTOR signaling cascade is central to cell survival, apoptosis, metabolism, motility, and angiogenesis [48].

In response to PI3K/AKT signaling activation, mTOR rapidly phosphorylates both downstream substrates, p70-S6K and 4E-BP1, the latter leading to release of EIF4E, resulting in initiation of translation. This pathway was also found to be up-regulated using tissue microarrays in CCA [27] and is a key pathway for CCA drug development [30]. mTOR can be inhibited by using the macrolide rapamycin. However, a subset of biliary cancers will be possibly resistant to mTOR inhibitors as the downstream activation bypass mTOR regulation. Therefore, in a study of gene expression comparing BTC and normal biliary epithelium identified two genes involving mTOR pathway, p70-S6K and EIF4E, which are differentially up-regulated in BTC so this study provides alternative downstream targets for inhibition [47].

In contrast, mTORC2 contains Rictor in place of Raptor so it phosphorylates a distinct set of substrates [49]. AKT/mTORC2 forms a positive feedback loop that AKT phosphorylates SIN1 at Tyrosine 86 which enhances mTORC2 kinase activity to phosphorylate and catalyse AKT(Serine 473) leading to AKT activation to control various cellular processes [50,51]. mTORC2 is tumorigenic and is reported to promote cancer via formation of lipids essential for growth and energy production in hepatocellular carcinoma model [52,53,54].

### 4.2. Upstream Regulation of mTOR in Cancers

#### 4.2.1. The Physiological Regulation of mTOR Pathway

PI3Ks are a family of intracellular signal transducers and regulate a crucial signal transduction system linking multiple receptors and oncogenes to many essential cellular functions including cell survival, proliferation, and differentiation [55]. Upon signals from various growth factors and cytokines stimulating receptor tyrosine kinases (RTKs) and G protein-coupled receptors (GPCRs), PI3Ks transduce the signals into intracellular messages via activating the serine/threonine kinase AKT followed by downstream effector pathways.

Several classes of PI3K kinases have been identified in mammalian cells, and only class I PI3K can function as second-messenger being implicated in oncogenesis. The class I PI3K kinase consists of two main subunits, p85 and p110, which mediate regulatory and catalytic activity of kinase respectively [56]. Three different genes, *PIK3CA*, *PIK3CB*, and *PIK3CD*, encode three specific p110 isoforms, p110*α*, *β*, and *δ*, respectively [57], and activating missense mutations of *PIK3CA* have been found as oncogenic in a variety of cancers [58]. *PIK3CB* mutation is rare but has been reported to be activating and oncogenic [59].

The PI3K kinases activated by RTKs phosphorylate the 3′-hydroxyl group of phosphatidylinositol (4,5)-bisphosphate (PIP2) to generate phosphatidylinositol (3,4,5)-trisphosphate (PIP3) [60], which is an important second messenger that transduces signals through AKT to downstream activators of cellular growth and survival [61]. PTEN is a phosphatase which negatively regulates PIP3 activity by dephosphorylation [62].

AMPK activity can be regulated by the cellular energy level through the balance in ATP/AMP ratio, so low ratio under nutrient deprivation can activate AMPK followed by mTOR inhibition via TSC1/2 activation [63]. p53 was reported as a substrate of AMPK which activates p53 phosphorylation on serine 15 required to initiate AMPK-dependent cell cycle arrest [64]. In addition, AMPK, TSC2, and PTEN were also regulated by p53 [65]. Furthermore, MAPKinase pathway activates ERK/RSK which regulate mTOR via TSC-2 suppression [66]. Therefore, those pathways are tightly regulated to affect mTOR activities leading to the balance of cell survival and death (Figure 1).

#### 4.2.2. Alterations of mTOR Pathway in Cancers

IGF1-R is a receptor on the cell surface and stimulated through the binding with IGF1 resulting in activation of PI3K/AKT/mTOR pathway. IGF1-R overexpression was reported to be associated with more aggressive phenotypes of cancer [67]. PTEN is a negative regulator of PI3K so is considered as a tumor suppressor in tumorigenesis [62]. Dysregulation of the above genes or proteins leads to mTOR activation resulting in tumor progression and survival in BTC.

*PIK3CA* mutations are commonly found in cancers such colon, breast, gastric, and brain cancers, but such mutations are rarely found in BTC [18,27] and are associated with poor prognosis [68]. Although not high rate of *PIK3CA* activating mutations, immunohistochemical evaluation of downstream *PIK3CA* targets EIF4E and 4E-BP1 suggests that additional mechanisms may play positively regulation in mTOR pathway in cancers. In addition, *PTEN* downregulation was reported to be associated with mTOR activation in BTC [69]. Expression profiling of BTC compared with normal biliary epithelium has identified upregulated AKT/mTOR signaling components, including the potential drug target IGF1-R [47]. Expression of IGF1-R and its ligands are seen in the majority of GBCs and metastases providing a targeted candidate for therapeutic strategies to interfering with IGF pathway [70]. Therefore, treating BTC cell lines with a small-molecule inhibitor of the IGF1-R was suggested and showed the efficacy of targeting this pathway [71].

## 5. mTOR Inhibitors

### 5.1. Rapalogs, First-Generation of mTOR Inhibitors

Rapalogs include rapamycin, also known as sirolimus, and its analogues such as everolimus, temsirolimus are all highly specific allosteric inhibitors of mTOR with the same mechanism of action [72,73]. Rapalogs binding to the intracellular protein FKBP12 forms a drug-protein complex. This FKBP12–rapalog complex binds to the FKBP12–rapamycin binding (FRB) domain of mTOR, which is located at just N-terminal next to the kinase domain [74,75]. Binding of FKBP12–rapalog complex to the FRB domain interferes the association of mTOR and Raptor in mTORC1 so that inhibits mTORC1 signaling within minutes at low doses of rapalogs. In contrast, higher doses or prolonged use of rapalogs can sequester mTOR from mTORC2 to block mTORC2 signaling [27]. Although rapalogs are highly specific to mTOR, it is well known that rapalogs can only partially inhibit the functions of mTORC1 [76,77]. For example, rapamycin highly inhibits S6K activity in all settings but does not inhibit 4E-BP1 which is also a direct substrate of mTORC1 [76]. Therefore, the sensitivity to rapalogs cannot determine whether the cellular processes are mTORC1-dependent or mTORC1-independent.

### 5.2. Second-Generation mTOR Inhibitors

For the limitations of rapalogs in mTOR inhibition, a number of second-generation mTOR inhibitors have been developed. Like most kinase inhibitors, second-generation mTOR inhibitors were designed to directly target the ATP-binding pocket of the mTOR kinase domain so these new generation mTOR inhibitors can inhibit both mTORC1 and mTORC2. The next important question is whether these compounds display superior anti-cancer activity via inhibition of both mTORC1 and mTORC2 and whether such treatments can be tolerated at the effective doses because of off-target effects on the evolutionarily related protein kinases [78]. Currently, several compounds such as AZD-2014, MLN0128 (INK128, TAK228), OSI-027, and GDC-0349 have been investigated in clinical trials to prove the clinical significance in cancer treatments. Furthermore, NVP-BEZ235, LY3023414, and PF-04691502 are dual PI3K/mTOR inhibitors and have been investigated in clinical trials.

AZD-2014, a dual mTORC1/2 inhibitor, showed superior activity than everolimus in vitro and in vivo in renal cell carcinoma [79] but demonstrate inferior efficacy in patients with renal cell carcinoma [80]. Therefore, although preliminary studies showed promising efficacy of dual mTOR inhibitors in various of cancers [81,82], the clinical significance should still be investigated in clinical trials to prove their activities in cancer treatment [83].

## 6. Sustained mTORC1/2 Signaling Activation as a Driver of Resistance to Anti-Cancer Treatment

Several studies in different cancer types have already shown that sustained mTORC1 signaling under certain targeted therapy is strongly associated with primary and acquired resistance to such treatment so mTORC1 inhibition seems to be an effective therapeutic strategy in combination with other targeted agents even the efficacy is limited as single-use [84,85]. mTORC1 activation has been also reported to be resistant to various anti-cancer treatments including chemotherapy, targeted therapy, and hormonal treatment. On the contrary, mTOR inhibition by rapalogs was shown sensitization to anti-cancer treatments [86].

mTORC2 activation and AKT phosphorylation have also been found to escape MAPKinase inhibition by sorafenib in CCA cells. Therefore, prevention of escape by suppressing mTORC2 activity may lead to promising new approaches in CCA therapy [87].

## 7. Preclinical Studies of mTOR Inhibitors in BTC

### 7.1. The Rationale of mTOR Inhibitors Alone or in Combination with Chemotherapeutic Agents in Cholangiocarcinoma

As discussed above, a number of genetic alterations directly or indirectly involving PI3K/AKT/mTOR activation were reported in advanced BTC [30]. In addition, gene expression profiling of invasive BTC has showed upregulation of downstream mediators in mTOR pathway, *RPS6K* and *EIF4E* as well as IGF1-R [32]. These genetic studies in BTC suggest mTOR plays an important role in invasive BTC, therefore, mTOR inhibitors targeting mTOR pathway could be considered as a reasonable therapeutic strategy.

Furthermore, in a preclinical study to investigate the functional role and mechanism of miR-199a-3p in the regulation of cisplatin sensitivities in CCA, Li et al. demonstrated that miR-199a-3p enhances cisplatin activity in CCA cell lines (GBC-SD and RBE) via both inhibiting the mTOR signaling pathway and decreasing the expression of MDR1. In this study, mTOR suppression by siRNA or miR-199a-3p potentiates cisplatin sensitivity of CCA cell lines indicating mTOR pathway regulates cisplatin activity in CCA although the exact mechanism is unclear [88].

Ling et al. found metformin increases AMPK phosphorylation and inhibits the activation of mTORC1 complex and can sensitize sorafenib, 5-FU, and As2O3 but not gemcitabine in cholangiocarcinoma cell lines (RBE and HCCC-9810) [89]. Wandee et al. found metformin sensitizes cisplatin in CCA cell lines (KKU-100 and KKU-452) via AMPK activation and AKT/mTOR/p70-S6K suppression [90]. Lyu et al. demonstrated Fyn was associated with AMPK/mTOR regulation [91] and was overexpressed in CCA cell lines. Furthermore, Fyn knockdown in CCA cell lines induces AMPK phosphorylation, followed by inhibiting downstream mTOR phosphorylation leading to inhibition of migration and invasion [92].

Above studies have shown mTOR pathway is crucial for regulation of tumor growth and sensitivities to anti-cancer drugs in CCA.

### 7.2. Preclinical Studies of Rapalogs in BTC

Everolimus exhibits in vitro multiple effects in a CCA cell line (RMCCA-1). Everolimus at low concentrations reduced in vitro invasion and migration and high concentrations exhibited cytotoxic effects such as suppression of cell proliferation and induction of apoptosis [93]. Everolimus was also found to inhibit the secretion of proinflammatory cytokines by cancer-associated myofibroblasts (CAFs) and inhibits proliferation of CCA cells (HuCCT1 and TFK1) at low concentrations [94]. Both studies confirmed the previous hypothesis that mTOR plays important role in CCA and mTOR inhibitors exert anticancer effects via mTOR inhibition. In addition, rapamycin was found to initiate AKT activation in CCA and inhibition of AKT by salubrinal potentiates the in vitro and in vivo efficacy of rapamycin in CCA both [95]. In terms of combination of rapalogs and cytotoxic agents, our group reported gemcitabine plus everolimus combination showed synergistic effect in the CCA cells in vitro and in vivo [96].

### 7.3. New Generation mTOR Inhibitors in BTC

Zhang et al. established a novel mouse model of intrahepatic CCA exhibiting activated AKT/mTOR cascade and found both mTORC1 and mTORC2 signalings are required for AKT/YapS127A-induced cholangiocarcinogenesis [97]. MLN0128, a second generation, ATP-competitive mTOR inhibitor, suppress cell growth and induce apoptosis in vitro and in vivo via suppression of both mTORC1 and mTORC2 signaling. An important finding in this study was that MLN0128 had better therapeutic efficacy than gemcitabine/oxaliplatin combination (one of the standard chemotherapy regimen) as well as everolimus in the treatment of AKT/YapS127A intrahepatic CCA model. In addition, the same group reported that the combination of palbociclib, a CDK4/6 inhibitor, and MLN0128 demonstrated a pronounced, synergistic growth inhibition in intrahepatic CCA cell lines and in AKT/YapS127A mice [98].

### 7.4. Dual PI3K/mTOR Inhibitors in BTC

New dual inhibitors targeted to PI3K/mTOR such as NVP-BEZ235, which exerts strong antiproliferative properties against primary cultures of intrahepatic CCA subtypes with differential drug sensitivity, have been developed [99]. In addition, our group identified both HSP90 overexpression and loss of PTEN were poor prognostic factors in patients with intrahepatic CCA. Thus, the combination of the HSP90 inhibitor (NVP-AUY922) and the PI3K/mTOR inhibitor (NVP-BEZ235) in CCA were evaluated and showed synergistic effects in vitro and in vivo. This combination not only inhibited the PI3K/AKT/mTOR pathway but also induced reactive oxygen species (ROS), which may enhance the vicious cycle of endoplasmic reticulum (ER) stress. Our data suggest the simultaneous targeting of the PI3K/mTOR and HSP pathways could be a novel and active therapeutic strategy for advanced CCA [100].

### 7.5. Other Indirect Inhibition of mTOR Pathway

VEGF can induce phosphorylation of both VEGFR1 and VEGFR2 but only VEGFR2 played a role in the promoting anti-apoptotic cell growth through activating a PI3K/AKT/mTOR signaling pathway. Apatinib, a VEGFR2-specific inhibitor, was reported to inhibit the anti-apoptosis induced by VEGF signaling, and promoted cell death in vitro and delayed tumor growth in vivo [101].

Besides direct inhibitors targeting mTOR, suppression of MAPKinase [102,103] or reactivation p53 [104,105] alone or in combination with mTOR inhibitors could be reasonably therapeutic strategies in advanced BTC.

## 8. mTOR Inhibitors in Clinical Setting

There are two settings for mTOR inhibitors used in the patients with advanced cholangiocarcinoma. Firstly, mTOR inhibitors could be used alone or in combination with other agents in the patients with advanced cholangiocarcinoma refractory to standard treatments. Secondly, mTOR inhibitors could be used in combination with standard treatment in patients with treatment-naïve advanced cholangiocarcinoma to investigate the possibly better response, progression-free survival and overall survival than conventional standard treatment. The combination of mTOR inhibitors with standard treatment aims to overcome resistance and potentiate the cytotoxicity of chemotherapy. Published clinical studies of mTOR inhibitors in advanced BTC were summarized in Table 1.

### 8.1. Clinical Studies of Everolimus in Advanced BTC

Bian et al. reported a 31-year-old male patient was diagnosed as stage IV intrahepatic CCA with *PIK3CA* mutation (E545G), which may result in activating mTOR pathway so patient received everolimus and achieved partial response (PR) after 2-month everolimus and at least 6-month PFS [106]. Larger series of everolimus in advanced BTC were studied. A phase I study reported that everolimus achieved 50% disease-control-rate (DCR) in a subgroup of 22 advanced BTC patients [107]. A phase II ITMO study in Italy enrolled 39 patients with previously chemotherapy-treated advanced BTC and the DCR was 44.7%, and the RR was 5.1%. Among two patients who experienced response, one patient showed a PR at 2 months and another patient showed a complete response (CR) sustained up to 8 months. The median PFS and OS were 3.2 (CI: 1.8–4.0) and 7.7 (CI: 5.5–13.2) months respectively [108]. In another phase II study to evaluate the activity of everolimus in 10 patients with *PIK3CA* amplification/mutation or *PTEN* loss refractory solid cancer, one patient with CCA with *PTEN* loss experienced disease control [109]. Recently, another phase II the RADiChol study published to evaluate the efficacy of everolimus as first-line treatment in treatment naïve advanced BTC, 27 patients enrolled showed DCR at 12 weeks was 48% and PFS and OS were 5.5 (2.2–10.0) and 9.5 (5.5–16.6) months, respectively [110]. Three (12%) of 25 patients evaluable for response experienced PR and 15 patients had stable diseases (SD). In addition, the authors performed immunohistochemistry (IHC) staining of PI3K/AKT/mTOR and found no association between IHC and clinical outcomes [110].

### 8.2. Clinical Studies of Sirolimus in Advanced BTC

In terms of other mTOR inhibitors, sirolimus, Rizell reported a cohort of sirolimus used in patients with hepatocellular carcinoma (*n* = 21) and iCCA (*n* = 9). Three (33%) of nine patients with intrahepatic CCA achieved SD after sirolimus treatment and others experienced progressive disease [111]. In a pilot study enrolling patients with PIK3CA mutant/amplified refractory solid cancer, sirolimus failed to demonstrate the clinical benefit in a patient with hilar cholangiocarcinoma (PIK3CA E545K mutation) who experienced disease progression following the second cycle of sirolimus with PFS of 0.9 months [112].

mTOR inhibitors alone, either sirolimus or everolimus showed some activity in advanced BTC with acceptable toxicities in treatment-naïve or pre-treated advanced BTC. The DCR (~50%) and survivals are compatible with the current standard of care. The use of mTOR inhibitors should be validated by larger randomized controlled trial (RCT) studies particularly in treatment naïve patients. For refractory BTC patients, mTOR inhibitors provide limited disease control which might benefit some patients whose BTCs are refractory to standard treatment.

The only published study investigating the combination of everolimus and chemotherapy was performed to determine the maximally tolerated dose (MTD) of different combinations [113]. The MTD for Cohort I of the two-drug combination was everolimus 5 mg on Monday/Wednesday/Friday and gemcitabine 800 mg/m^2^. Cohort II was to determine the MTC when cisplatin was added in everolimus/gemcitabine as a three-drug combination and cohort III was evaluation the activity of everolimus 5 mg on Monday/Wednesday/Friday, gemcitabine 600 mg/m^2^, cisplatin 12.5 mg/m^2^, 60% of 10 CCA and GBC carcinoma in cohort 3 experienced SD. The hematological DLT (mainly thrombocytopenia) limited the dosage used in three-drug combination and resulted in limited response rate. However, everolimus/gemcitabine could be an interesting regimen which demonstrated 2 CRs in this two-drug combination.

### 8.3. Clinical Studies of New Generation mTOR Inhibitors in Advanced BTC

Although new generation mTOR inhibitors and dual PI3K/mTOR inhibitors targeting both mTORC1 and mTORC2 showed anticancer activities in BTC (discussed in Section 8.2 and Section 8.3), no clinical trials of these agents in advanced BTC were reported. All of these compounds are being investigated under early clinical trials to evaluate the efficacy in various refractory solid or hematologic cancers. Therefore, more and more results of new generation mTOR inhibitors will be released and published in the near future.

## 9. Summary of mTOR Inhibitors in BTC

In conclusion, mTOR signaling pathway connecting with several other pathways and networks regulates cancer proliferation and progression. Activation of mTOR is associated with drug-resistance in BTC. Rapalogs partially inhibit mTORC1 and exhibit anti-cancer activity. Rapalogs in clinical trials demonstrate some activity in patients with advanced BTC. New-generation mTOR inhibitors against ATP-binding pocket inhibit both TORC1 and TORC2 and demonstrate more potent anti-tumor effects in vitro and in vivo. Prospective clinical trials are warranted to prove its efficacy in patients with advanced BTC.

## Figures and Tables

**Figure 1 ijms-20-00500-f001:**
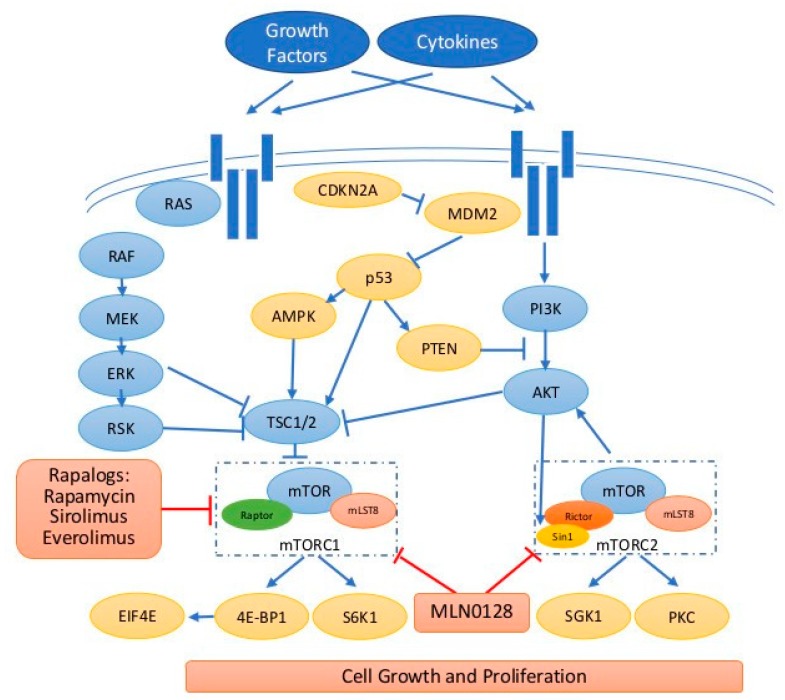
The signaling transduction of mTOR pathway. Extracellular signals such as growth factors and cytokines binding to the receptors stimulate RAS/RAF/MEK/ERK and PI3K/AKT/mTOR caspases. mTOR exists in two functionally and structurally distinct complexes, mTORC1 and mTORC2. Both mTORC1/2 contain different core components so they phosphorylate a distinct set of substrates and exhibit distinct function. PTEN is a negative regulator for PI3K/AKT. In addition, ERK/RSK, AMPK, and p53 regulate mTORC1 through TSC2 regulation. Rapalogs mainly inhibit mTORC1 and new-generation mTOR inhibitors such as MLN0128 inhibitor both mTORC1/2. Blue t-bar indicates inhibition, blue arrow indicates stimulation/activation, red t-bar indicates inhibition by drugs, and dashed square indicates mTOR1/2 complexes.

**Table 1 ijms-20-00500-t001:** Summary of published data regarding mTOR inhibitors in advanced BTC.

Compound(s)	Phase	Patients	Response	Survival
Everolimus, 1 L [106]	Case report	iCCA (*n* = 1) with PIK3CA mutation	PR	PFS > 6 m
Everolimus [107]	Phase I	Advanced BTC (*n* = 22)	DCR: 50% (11/22)	NA
Everolimus (>2 L) [108]	Phase II	Advanced BTC (*n* = 39)	DCR: 44.7% RR: 5.1% (including 1 CR)	mPFS: 3.2 m (1.8–4.0) mOS: 7.7 m (5.5–13.2)
Everolimus [109]	Phase II	CCA (*n* = 1), PTEN loss	SD	NA
Everolimus (1 L) [110]	Phase II	Advanced BTC (*n* = 27)	DCR at 12 weeks: 48% PR: 12% (3/25) SD: 60% (15/25)	mPFS: 5.5 m (2.2–10.0) mOS: 9.5 m (5.5–16.6)
Sirolimus [111]	Phase II	iCCA (*n* = 9)	SD: 33% (3/9) PD: 67% (6/9)	mOS:7 (2.6–35)
Sirolimus [112]	Phase II	hilar CCA (*n* = 1) with PIK3CA mutation	PD	PFS: 0.9 m
Everolimus, gemcitabine, cisplatin (1 L) [113]	Phase I	Cohort III, CCA and GBC (*n* = 10)	SD: 60% (6/10) PD: 40% (4/10)	NA

1 L, first line; 2 L, second line; iCCA, intrahepatic cholangiocarcinoma; GBC, gallbladder cancer; BTC, biliary tract cancer; CR, complete response; PR, partial response; SD, stable disease; PD, progressive disease; DCR, disease control rate; mPFS, median progression-free survival; mOS, median overall survival.

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
