# Peer review of "mTOR Inhibitors in Advanced Biliary Tract Cancers"

_ijms, 2019, doi:10.3390/ijms20030500_

Round 1

Reviewer 1 Report

COMMENTS TO AUTHORS:

Title: mTOR inhibitors in advanced biliary tract cancers

This review is focus on mTOR inhibitors and their potential therapeutical use in advanced biliary tract cancer. The authors introduces BTCs and their currently treatment that is not totally effective. They propose to use mTORC1 inhibitors against BTC, based on the observed activity increase of the mTORC1 pathway in these tumors. Finally, the authors explain some clinical trials using mTOR inhibitors.

In PubMed there are some review related with this subject published in high impact journals.  (Bhat, M., Sonenberg, N. and Gores, G. J. 2013-Hepatology, 58: 810-818. Corti, F; Nichetti,  F; Raimondi, A et al. 2019-Cancer Treatment Reviews,72:45-55), but the structure of this manuscript and the analysis of the subject is complementary to these publications.

The structure of this manuscript is correct and the authors give relevant information about preclinical and clinical trial using mTOR inhibitors in biliary tract cancer treatment.

However there are minor errors that if corrected can improve the text:

-          Some titles lead to confusion: mTOR its complexes and downstream regulations in BTC or Upstream regulation of mTOR in BTC. Both section describe upstream and downstream elements in mTORC pathway that are common in all cellular types, are not specific from BTC. In my opinion, the authors could change the title for more appropriate one. In addition, in line 126 the section title is not numbered.

-          I could not check the figure 1 because is not in the PDF.

-          In sections 3, 4 some words are in blue color and underlined.

-          Line 164-165 RTKs are tyrosine kinase receptors not necessarily cytokines receptors.

-          Line 189 IFG1-R must be change for IGF1-R.

-          In the section 5.2: Second generation mTOR inhibitors, the authors enumerates some compounds such as: MLN0128, OSI-027, GDC-0349 and others PI3K/mTOR inhibitors, that have been tested in clinical trials in cancer, but they do not make clear if this compounds have showed some efficiency against biliary tract cancer. In addition, they do not expose a critical opinion about their possible application in treatment of BTC. In my opinion, just a list of tested compounds without a critique opinion of them, adds nothing to a scientific review.

-          Sections 7 and 8 are a little disorganized; perhaps the text could be improved if they were merged in a single section that contained BTC and CCA results.

-          Lines 273-277, section 8.1 page 6; lines 292-296, section 8.3 page 7: Too long sentences need English revision.

Author Response

18, January, 2019

Olivier Dormond

Guest Editor of International Journal of Molecular Sciences

Re: ijms-431449

“mTOR inhibitors in advanced biliary tract cancers” By: Chao-En Wu, Ming-Huang Chen, Chun-Nan Yeh

Dear Professor Dormond

We are pleased to be informed that the above manuscript has been reviewed. All the valuable comments from the reviewers are sound and appreciated. We have revised the manuscript to comply with changes requested by the reviewers. Attached please find the revised version of the manuscript and a point to point response to the requested revisions. The changes made are displayed in the attached revised manuscript file using the Word tracking system.

In addition, both reviewers mentioned they cannot review the figure 1 in the manuscript which is possibly missed after your formatting. Please help us to correct it.

We deeply believe the revised version will fulfil the requirements of publication in the journal of “International Journal of Molecular Sciences”.

Yours sincerely,

Chun-Nan Yeh

Department of Surgery and Liver Research Center

Chang Gung Memorial Hospital

Chang Gung University

Taoyuan 333, Taiwan

Phone: 886-33281200, extension 3219

Fax: 886-33285818

E-mail: yehchunnan@gmail.com

Point-to-Point response

Response to Reviewer 1:

 Comments and Suggestions for Authors

COMMENTS TO AUTHORS:

Title: mTOR inhibitors in advanced biliary tract cancers

This review is focus on mTOR inhibitors and their potential therapeutical use in advanced biliary tract cancer. The authors introduces BTCs and their currently treatment that is not totally effective. They propose to use mTORC1 inhibitors against BTC, based on the observed activity increase of the mTORC1 pathway in these tumors. Finally, the authors explain some clinical trials using mTOR inhibitors.

In PubMed there are some review related with this subject published in high impact journals.  (Bhat, M., Sonenberg, N. and Gores, G. J. 2013-Hepatology, 58: 810-818. Corti, F; Nichetti,  F; Raimondi, A et al. 2019-Cancer Treatment Reviews,72:45-55), but the structure of this manuscript and the analysis of the subject is complementary to these publications.

The structure of this manuscript is correct and the authors give relevant information about preclinical and clinical trial using mTOR inhibitors in biliary tract cancer treatment.

However there are minor errors that if corrected can improve the text:

-          Some titles lead to confusion: mTOR its complexes and downstream regulations in BTC or Upstream regulation of mTOR in BTC. Both section describe upstream and downstream elements in mTORC pathway that are common in all cellular types, are not specific from BTC. In my opinion, the authors could change the title for more appropriate one. In addition, in line 126 the section title is not numbered.

Reply: Thanks for your valuable comments. We have replaced “BTC” by “cancers” in those titles in revised manuscript. We have also numbered the title in line 126.

I could not check the figure 1 because is not in the PDF.

Reply: We submitted manuscript along with the figure 1 but we don’t have any ideas about why the figure 1 is not shown in the PDF file. We will ask the editorial board to add this figure in the revised manuscript.

Figure 1. The signaling transduction of mTOR pathway. Extracellular signals such as growth factors and cytokines binding to the receptors stimulate RAS/RAF/MEK/ERK and PI3K/AKT/mTOR caspases. mTOR exists in two functionally and structurally distinct complexes, mTOCR1 and mTORC2. Both mTORC1/2 contain different core components so they phosphorylate a distinct set of substrates and exhibit distinct function. PTEN is a negative regulator for PI3K/AKT. In addition, ERK/RSK, AMPK and p53 regulate mTORC1 through TSC2 regulation. Rapalogs mainly inhibit mTORC1 and new- generation mTOR inhibitors such as MLN0128 inhibitor both mTORC1/2.

In sections 3, 4 some words are in blue color and underlined.

Reply: Those didn’t happen in our original manuscript we submitted to the journal so those might came from some errors during formatting process. We have corrected those in the revised manuscript.

Line 189 IFG1-R must be change for IGF1-R.

Reply: We have changed it in the revised manuscript.

In the section 5.2: Second generation mTOR inhibitors, the authors enumerates some compounds such as: MLN0128, OSI-027, GDC-0349 and others PI3K/mTOR inhibitors, that have been tested in clinical trials in cancer, but they do not make clear if this compounds have showed some efficiency against biliary tract cancer. In addition, they do not expose a critical opinion about their possible application in treatment of BTC. In my opinion, just a list of tested compounds without a critique opinion of them, adds nothing to a scientific review.

Reply: In terms of second generation mTOR inhibitors, there are only some preclinical studies in BTC which were mentioned in the sections 8.2 and 8.3 in the manuscript. The clinical trials of such novel compounds in BTC are lacking and we have added this in the revised manuscript (new section 8.3).

Sections 7 and 8 are a little disorganized; perhaps the text could be improved if they were merged in a single section that contained BTC and CCA results.

Reply: Thanks for your suggestion. We have merged two sections in the revised manuscript.

Lines 273-277, section 8.1 page 6; lines 292-296, section 8.3 page 7: Too long sentences need English revision.

Reply: We have revised them in the revised version to make it easy to read.

Reviewer 2 Report

The manuscript titled " mTOR inhibitors in advanced biliary tract cancers" reviews the (i) limitations of current treatment in advanced biliary tract cancers (ii)  the importance of mTOR signaling (iii) possible role of mTOR inhibitors as a future treatment in BTC. The review is well written and apt for the International Journal of Molecular sciences. I have few minor comments that could be addressed.

1.     A schematic or a figure depicting the molecular mechanisms of some of the mTOR inhibitors. Especially the new generation inhibitors against the ATP-binding pocket targeting TORC1 and TORC2.

2.     Line 124, manuscript refers to a figure. However, I could not find in the submitted manuscript.

3.     Spacing before the reference citation is required. This could be fixed editorially.

4.     Some of the text in the manuscript is either highlighted (in gray) or in bold. The significance of it is not clear.

5.     Line 370. "Reference" to "References"

6.     Reference formatting should meet the journal guidelines.

Author Response

18, January, 2019

Olivier Dormond

Guest Editor of International Journal of Molecular Sciences

Re: ijms-431449

“mTOR inhibitors in advanced biliary tract cancers” By: Chao-En Wu, Ming-Huang Chen, Chun-Nan Yeh

Dear Professor Dormond

We are pleased to be informed that the above manuscript has been reviewed. All the valuable comments from the reviewers are sound and appreciated. We have revised the manuscript to comply with changes requested by the reviewers. Attached please find the revised version of the manuscript and a point to point response to the requested revisions. The changes made are displayed in the attached revised manuscript file using the Word tracking system.

In addition, both reviewers mentioned they cannot review the figure 1 in the manuscript which is possibly missed after your formatting. Please help us to correct it.

We deeply believe the revised version will fulfil the requirements of publication in the journal of “International Journal of Molecular Sciences”.

Yours sincerely,

Chun-Nan Yeh

Department of Surgery and Liver Research Center

Chang Gung Memorial Hospital

Chang Gung University

Taoyuan 333, Taiwan

Phone: 886-33281200, extension 3219

Fax: 886-33285818

E-mail: yehchunnan@gmail.com

Response to Reviewer 2

Comments and Suggestions for Authors

The manuscript titled " mTOR inhibitors in advanced biliary tract cancers" reviews the (i) limitations of current treatment in advanced biliary tract cancers (ii)  the importance of mTOR signaling (iii) possible role of mTOR inhibitors as a future treatment in BTC. The review is well written and apt for the International Journal of Molecular sciences. I have few minor comments that could be addressed.

1.     A schematic or a figure depicting the molecular mechanisms of some of the mTOR inhibitors. Especially the new generation inhibitors against the ATP-binding pocket targeting TORC1 and TORC2.

Reply: We submitted manuscript along with the figure 1 but we don’t have any ideas about why the figure 1 is not shown in the PDF file. This figure can depict the molecular mechanism of mTOR inhibitors.

2.     Line 124, manuscript refers to a figure. However, I could not find in the submitted manuscript.

Reply: We submitted manuscript along with the figure 1 but we don’t have any ideas about why the figure 1 is not shown in the PDF file. We will ask the editorial board to add this figure in the revised manuscript.

Figure 1. The signaling transduction of mTOR pathway. Extracellular signals such as growth factors and cytokines binding to the receptors stimulate RAS/RAF/MEK/ERK and PI3K/AKT/mTOR caspases. mTOR exists in two functionally and structurally distinct complexes, mTOCR1 and mTORC2. Both mTORC1/2 contain different core components so they phosphorylate a distinct set of substrates and exhibit distinct function. PTEN is a negative regulator for PI3K/AKT. In addition, ERK/RSK, AMPK and p53 regulate mTORC1 through TSC2 regulation. Rapalogs mainly inhibit mTORC1 and new- generation mTOR inhibitors such as MLN0128 inhibitor both mTORC1/2.

3.     Spacing before the reference citation is required. This could be fixed editorially.

Reply: Thanks for your comments. We have spaced them in the revised manuscript.

4.     Some of the text in the manuscript is either highlighted (in gray) or in bold. The significance of it is not clear.

Reply: Those highlights did not appear in the original manuscript so some errors happened during formatting. We have tried to fix it in the revised manuscript.

5.     Line 370. "Reference" to "References"

Reply: We have changed it in the revised manuscript.

6.     Reference formatting should meet the journal guidelines.

Reply: We have changed them to meet the journal guidelines in the revised manuscript

Round 2

Reviewer 1 Report

Dear authors:

The manuscript has improved. The authors have answered all the questions raised  and have modified what has been proposed. 

In this version I could see the figure 1 and is appropriate and correct.